# Kittens That Nurse 7 Weeks or Longer Are Less Likely to Become Overweight Adult Cats

**DOI:** 10.3390/ani11123434

**Published:** 2021-12-01

**Authors:** Denise van Lent, Johannes C. M. Vernooij, Marcellina M. Stolting, Ronald Jan Corbee

**Affiliations:** 1Lekker in je Vacht, Feline Behavioral Health and Welfare Centre, Secretaris Runsinkbrink 6, 2731 AG Benthuizen, The Netherlands; 2Department of Population Health Sciences, Faculty of Veterinary Medicine, Utrecht University, Yalelaan 7, 3584 CL Utrecht, The Netherlands; j.c.m.vernooij@uu.nl; 3Cat Behaviour Consultancy, P.O. Box 11375, 1001 GJ Amsterdam, The Netherlands; info@kattengedragstherapie.nl; 4Department of Clinical Sciences of Companion Animals, Faculty of Veterinary Medicine, Utrecht University, Yalelaan 108, 3584 CM Utrecht, The Netherlands; r.j.corbee@uu.nl

**Keywords:** leptin, suckling, weaning, feline obesity, welfare, risk factors, nursing, overweight

## Abstract

**Simple Summary:**

Most pet cats are separated from the queen during the suckling period and before they are fully weaned. Weaning is characterized by rapid growth and development of the kitten and marks the transition from a fully milk-based diet to a solid-food diet. Early-life feeding is known to influence the development of the digestion system, eating behavior, and dietary patterns. The aim of this study was to examine the effects of the suckling period length (SPL) on adult weight status and whether a deficiency during a critical period of development—as a short SPL can be considered - may influence body weight regulation, control of appetite and energy expenditure. Our findings show that a shorter SPL increases the risk of overweight in cats. The odds for overweight was three times lower in cats with a SPL > 6 weeks (OR = 0.33, 95% CI = 0.10–0.99). The suckling period length (SPL) could be an easy modifiable risk factor in the primary prevention of overweight in cats.

**Abstract:**

The aim of this study was to examine the effect of the suckling period length (SPL) on weight status among adult cats while taking into account putative risk factors. To this end, the body fat percentage of 69 client-owned cats was determined. A body fat percentage of >30% was used for overweight classification. Cat owners were interviewed using a standardised questionnaire to collect information about the SPL, age, breed, sex, feeding amount and frequency, daily playing and outdoor access. SPL was categorized into four groups (0–6, 7–11, 12–16, 17–24 weeks). Logistic regression was used to estimate the association between overweight and SPL after adjusting for identified risk factors. Of the 69 cats, 37 were overweight. The odds for overweight was three times lower in cats with a SPL > 6 weeks (OR = 0.33, 95% CI = 0.10−0.99). This study identified a possible novel, modifiable early life risk factor for overweight in cats; the SPL. The results of this study indicate that allowing cats to nurse longer than 12 weeks might be a simple intervention to improve cat health and welfare.

## 1. Introduction 

Overweight in cats is of increasing concern as the prevalence is high (45%) and extra body fat increases the risk of diseases such as diabetes mellitus, cancer, arthritis and lower urinary tract disease [1,2,3]. In addition to these adverse effects on health, quality of life is decreased in dogs by limiting vitality, causing emotional disturbance and pain [4]. Furthermore, in dogs, cats and humans obesity has been associated with a shorter lifespan [5].

It is important to identify all risk factors to prevent overweight from occurring, as successful weight loss and subsequent weight management is difficult to achieve [6].

There is increased awareness for early life nutrition, because in this period dietary habits are established and metabolic pathways are being programmed [7]. Nutritional management of a new-born can have lifelong effects on many systems [8]. In general, the Infant’s first nutritional experiences after birth are through its mother’s milk. 

A number of human studies aimed at obesity prevention, investigated the long-term effects of breastfeeding on overweight [9,10]. Meta-analyses provided sufficient evidence for a dose-response protective effect of the duration of breastfeeding in children. [11,12,13,14]. The underlying biological mechanisms for the protective effect of prolonged breastfeeding are based on shaping appropriate eating behaviours and gut-microbiome, taste preferences and the development of self-regulatory abilities during feeding [9,15,16,17]. The appetite controlling hormone leptin, present in breastmilk is emerging as a key mediator in postnatal programming of a healthy adult phenotype [18].

Leptin intake during the suckling period protects against obesity in adult rats – even when fed a high fat diet – because it improves the metabolic response [19]. Oral treatment of obese human patients with low doses of leptin (0.01–0.04 mg/kg) has been effective in decreasing BMI [20]. Ingesting enough leptin through breast milk has been proposed as a prerequisite to ensure that the systems that regulate fat accumulation and ultimately determine body condition can function properly from the early stages of development [21].

In most mammals, a new born is exclusively fed breast milk during the suckling period. Mothers’ milk is replaced progressively by the adult diet which contains less fat and more carbohydrates during weaning [22]. In nature, weaning is a gradual process whereby suckling decreases as a function of increasing age as well as milk and food availability [23]. Data on how long free-living kittens are nursed for precisely are inconclusive. Some studies report 8–10 weeks as common, while others report that (intermittent) suckling may continue 3–5 months up to a year [24,25,26,27].

For domestic animals such as cats, weaning tends to be a rather abrupt and stressful event at an early age that coincides with the relocation to a new home, sudden maternal separation and dietary changes. Seven to eight weeks is a common point at which domestic shorthair kittens are taken away from the lactating queen [28]. In contrast, (Dutch) pedigreed kittens are not rehomed before 12–13 weeks of age due to pedigree registry regulations.

The suckling period length (SPL) could be an easy modifiable risk factor in the primary prevention of overweight in cats. The aim of this study was therefore to examine the effect of the SPL on adult weight status, while taking into account putative risk factors.

## 2. Materials and Methods

A total of 77 cat owners responded to an email that was sent out to clients of 3 veterinary practices requesting participation in the study, of which 69 cats met the inclusion criteria (Table 1).

In this study, SPL was defined as: “the amount of time during which the queen and her litter are housed as such, that nursing can continue as long as is mutually desired”.

Cat owners were informed that we were recruiting healthy, adult cats with a known SPL for a study focusing on a possible association between SPL and the development of overweight in adult life. All responding cat owners were send an additional email with a brief description of the study protocol.

The cats of interested cat owners were screened for inclusion criteria (Table 1).

When these criteria were met, an appointment was scheduled at one of the three participating practices. At the appointment, the questionnaire was filled in during a face to face interview, and the cat’s body fat mass was determined. Participation was completely voluntary and participants could withdraw from the study at any given time.

Prior to body mass assessment, the cats were free to acclimatize for 20 min. During this time, the cat owner answered questions in a face-to-face interview taken by a behavioural biologist (DvL). The questionnaire was pre-tested in a pilot study with a sample of five persons that were not part of the study population and included seven sections (cat description, SPL, dietary factors, general health, feeding regimen, and activity level).

Owners were asked for how long the cat was able to suckle as a kitten. The SPL was categorized into four groups: 0–6 weeks, 7–11 weeks, 12–16 weeks and >16 weeks, based on common adoption procedures. The age at which most domestic shorthair kittens are put up for adoption is 7–8 weeks, due to the Dutch legal separation age (7 weeks = 49 days) [29], whereas the age at which most purebred kittens are adopted out is 12–13 weeks due to the regulations of cat breeders associations [30,31]. Pedigree breeders are required to have completed the kittens’ first vaccination course before they can go to their new homes.

Only pedigreed cats with their ancestry recorded by a cat registry were classified as “purebred”. The study cats were divided into four age (measured in units of 0.5 year) categories (≥12 months–3 years, 3.5–5 years, 5.5–7 years and 7.5–12 years).

Cat owners answered questions about their care practices (lifestyle management of their cats) with regard to feeding regime and the use of the information on cat food labels (collected in accordance with Öhlund (2018) [5] and Naughton et al. (2021) [32].

Dietary factors included questions about the amount of food the owner fed in comparison to what is recommended by these pet food labels (less than, more than or the recommended amount of cat food). With regard to the feeding frequency, cats were either meal-fed (1–2 or 3–4 meals per day) or fed as much and as often as desired (ad libitum). Questions about the cat’s living environment were categorised as either “being able to roam freely outside” or “living indoors with at most limited outdoor access such as a patio, yard/balcony with a secure fence” (outdoor access yes/no), and exercise in the form of daily playing with the owner (yes/no).

A validated feline Body Mass Index (fBMI) was calculated from measured hind leg length and rib cage circumference (in cm) according to Hawthorne & Butterwick (2000) [33]. The cat owner was asked to loosely hold the cat in a standing position at the examination table. Cats were never restrained. Cats that would sit were manipulated into a standing position by offering a treat. The fBMI was chosen over BCS to reduce the subjective impact through the examiner [34,35]. The fBMI provides an indicator for Body Fat (BF) percentage. For statistical analysis, weight status at adulthood (≥12 months to 12.5 years) served as binary dependent variable and was classified into overweight (BF > 30%) or non-overweight (BF 11–30%) [36].

Data were entered in Excel and analysed with R statistical software [37]. Univariable analysis by binary logistic regression was performed as a means of identifying putative individual factors affecting the dependent variable, overweight. These factors were chosen on the basis of previously reported associations with feline obesity. SPL was the main independent variable. Age and breed were considered as possible confounders and sex, food amount and frequency, daily playing, and outdoor access were added as covariates. Results for the association of weight status with each of the respective risk factors were presented as univariable odds ratios (ORs) and 95% confidence intervals (CIs) with accompanying cross tables. The various factors were also tested for association with breed by Fisher’s exact test.

We used multivariable logistic regression to analyse which of the independent variables are associated with outcome overweight. All variables were entered in the model which was optimised using a backward stepwise approach based on Akaike’s Information Criteria (AIC). Adjusted ORs with 95% confidence intervals of the final model were computed and presented in the results. For sensitivity analysis the univariable models were also applied to a subset of only the domestic shorthair cats (*n* = 49). The final model of the full data was also applied on the subset as the data are too sparse to run the full model.

## 3. Results

### 3.1. Univariable Analysis

In this retrospective cohort study, 77 cat owners responded to an email sent by three Dutch veterinary practices. One cat was excluded from the study because he was underweight and seven cats were excluded from the study because they did not meet the age criteria (i.e. one cat was too old, and 6 cats were too young). Data of a total of 69 healthy adult cats met the inclusion criteria. The information is summarized in Table 2.

From the 69 cats, 34 cats were classified as overweight or obese, of which 27 were overweight (BF > 30–45%), and 7 obese (BF > 45%). The percentage of male cats (58.0%) was higher than that of female cats (42.0%). All but one animal were neutered. Mean age at measurement was 5.3 years (range ≥12 months–12 years). Of the 69 cats, 49 were domestic shorthairs and 20 purebred cats from nine different breeds (see Appendix A). Among purebred cats, Norwegian Forest cats (9/20) were the most frequently seen breed. Feeding the cat 1–2 meals per day or ad libitum, were the most popular choices in the whole group. Most cat owners (42/69) did not play with their cat on a daily basis.

The association between respective SPL categories and overweight showed a decreased risk for overweight with prolonged SPL (Table 2). In cats with a SPL between 0 and 6 weeks 68% were overweight in contrast to cats with longer SPL (33 to 48%). The odds ratio for overweight in cats that have a SPL > 6 weeks was 0.33 (95% CI = 0.10–0.99). Cats with age over 5 years and over 7 years showed a higher risk to be overweight (83% and 59%, respectively) than younger cats (28 to 41%). Purebred cats were less likely to be overweight (20%), compared to domestic short hair cats (61%). Other factors did not show an association with being overweight.

### 3.2. Multivariable Analysis

In the final model of the multivariable analysis, adjusted odds ratios for SPL and age were estimated. Age stayed in the final model while SPL remained in the model to answer the research question.

A longer SPL tended to decrease (2–4 fold lower) the likelihood to be overweight (Table 3) but were not significantly different from the shortest SPL category. Odds for overweight increased with age in cats older than 5 years of age compared to young adults.

The variables SPL and breed are entangled because on average, domestic shorthairs suckle for a considerable shorter period of time (8 weeks) than purebreds (16 weeks) (Table 4).

Several management factors were also associated with breed (Table 4): SPL, overweight status, age, number of meals per day, and outdoor access, respectively. Hardly any purebred cat was separated from the queen before 12 weeks, in contrast to the domestic shorthair cats. Purebred cats were less likely to be overweight, more often fed ad libitum, and less often had free outdoor access compared to domestic shorthair cats.

In Appendix A the results are presented of a similar analysis applied to the data of domestic short hair cats only, as the number of purebred cats is quite small. The results of these analyses show similar results as presented in Table 2 and Table 3 except that the estimates are less precise (wider 95% confidence interval).

## 4. Discussion

The aim of this study was therefore to examine the effect of the SPL on adult weight status, while taking into account putative risk factors. This retrospective cohort study which included 69 pet-cats, revealed cats that suckled longer than 6 weeks were about three times less likely to be overweight than cats that suckled 6 weeks or less. A prolonged SPL was thus associated with a lower prevalence of overweight irrespective of other risk factors.

Feline obesity is a major welfare concern and studies suggest that anywhere from 11.5% to 63% cats are overweight [38]. The combined prevalence of overweight and obesity found in this study (50.7%) is comparable to what has been previously reported in the Netherlands (50%) and Japan (56%), but higher than in the United States (35.1%), New Zealand (27.4%), Australia (32.8%) and France (26.8%) [39,40]. Housing, neutering and feeding practices may differ among countries leading to varying risk factors per geographic area [38].

Overweight is a multifactorial disease resulting from a combination of causes and individual factors and since weight loss and long-term maintenance of healthy weight is proven to be difficult for cat owners, it is important to identify all preventable risk factors.

A deficiency during a critical period of development—as a short SPL can be considered—may influence body weight regulation, control of appetite and energy expenditure according to Palou [15]. A more recent study suggested that adequate leptin intake during lactation might play a crucial role in the postnatal programming of future body weight and metabolic health [21]. More specific, leptin may play a key role in shaping the systems that control fat accumulation and body composition. Thus, cats with a short suckling period might be deprived of sufficient leptin ingested as a component of breast milk, leading to disturbances in these systems. While the adequate leptin intake of cats with a longer SPL may be protective against overweight. This study shows that the risk of overweight is lower with increased suckling period and suggests that a SPL of at least 7 weeks is sufficient to benefit from the protective effect.

To optimize the well-being of kittens as well as adult cats, it is essential to consider the weaning age and practices used as well as their relationship to developmental processes. In nature, weaning is a rather gradual transition from mothers’ milk to solid food. An important component of the process involves the kitten’s independence of the mother cat. By the time her kittens are about 6 to 8 weeks old, the queen starts with gradually increasing periods of separation. Over time, she will interact less with the kittens on a regular basis to teach them to be independent.

For pet cats, being adopted out to a new owner, automatically means being separated from the mother and the abrupt termination of the suckling period. At which point the suckling period is terminated, largely depends on whether the pet cat is a purebred or domestic shorthair (in the Netherlands). On the one hand, the legal separation age of Dutch kittens in general is regulated by animal welfare legislation, which states that kittens may not be separated from the mother before the age of 7 weeks. In Finland, the minimum separation age is 12 weeks, while in countries such as the United States, 8 weeks is more common [41].

On the other hand, the separation age of pedigreed purebred kittens is being regulated by cat breeding guidelines that dictate that kittens are only allowed to leave the nest at 12–13 weeks. Thus, in the case of pedigreed cats the breeders decide at what age their kittens leave the nest. In the case of domestic shorthairs it may be a private owner, a shelter or a rescue.

Our study reveals that almost all pedigreed purebred cats have a longer SPL (≥12 weeks) compared to domestic shorthairs (SPL < 12 weeks). Only one purebred cat had a SPL of 0 weeks because the mother cat died just after giving birth to the kitten.

Being domestic shorthair has been identified as a risk factor by several studies while being purebred was protective against overweight [5,42,43] but the underlying cause for this difference has never been identified. The current study suggests that overweight in cats might be associated with a shorter suckling period of domestic shorthairs.

As cat owners play an important role in their cats’ weight-related behaviours, cat management might facilitate or impede a healthy weight. Toribio (2009) suggested that purebred cat owners are more likely to seek veterinary attention and might be better informed concerning healthy eating habits and physical activity to maintain a healthy weight [44]. A recent study found that purebred owners were more likely to conduct research prior to acquisition of their cat compared to domestic shorthair owners [45]. Teng (2017) determined that purebred cats are more often chipped and neutered, suggesting that purebred owners’ sense of responsibility may be higher [46]. Although a higher sense of responsibility and being better informed potentially lead to better care practices, it is currently unknown whether and how these factors actually contribute to leaner cats. Despite the fact that these management factors were different between purebred and domestic shorthair cats (Table 3), this did not result in significant differences in the ORs (Table 2 and Table 4, Appendix A). Haring et al. have recently shown that a genetic component contributes to the development of overweight in cats [47]. Certain breeds (Birman, Abyssinian, Cornish Rex, Devon Rex, Oriental Shorthair, Siamese and Sphynx) may have biologic factors that protect against obesity [39,48,49]. In our study, 4% belonged to one of the breeds with 2 Siamese, 1 Birman and 1 Cornish Rex. Therefore, it seems unlikely that the protective effect of SPL would be due to a predominance of breeds with decreased risk of obesity, since more breeds were included with an increased risk, as 15.9% belonged to the breeds that might be at risk for overweight, with 9 Norwegian Forest cats, 1 British Shorthair, and 1 Persian cat. Breed standards of these breeds might encourage breeding cats that are prone to obesity [5,39,48].

Nutritional management is also an important management factor to consider, because feeding practices inevitably influence the cats food consumption and ultimately, body weight. In contrast to what one might expect, the odds for overweight were nearly halved with regard to the food amount when cats were fed more than recommended or ad libitum, although the number of cats per group was small. This result is similar to the study of Wall [48] that found “indulgent feeding” such as treat feeding, offering multiple different meals, offering special foods on special occasions and allowing the cat to choose when to eat to be protective against overweight. Robertson [43] found a similar result, as in that cats with access to food throughout the day were leaner. This is in contrast to some previous studies that identified both “ad lib feeding”, as well as being “meal fed twice a day” as risk factors for obesity [50,51]. Other studies found no association between feeding regimen and overweight [5,52].

Several studies that focused on the natural feeding behaviours of cats, suggest that feeding cats once or twice a day from cat food bowls may not address the domestic cat’s need for eating frequent small meals per day [53]. Both in the wild and when given free access to food (ad libitum), cats will nibble throughout the day and meal frequency can be as high as 20 small eating moments per day [54,55]. A recent study found that cats eat fewer but larger meals in a shorter time and eat faster in response to calorie restriction [56]. So it may be that frequent feeding promotes satiety, normal eating behaviour and smaller portions per meal, which contributes to maintaining a healthy weight. An alternative explanation could be that mostly finicky, picky cats are fed more treats or extra meals to stimulate eating. Therefore, ad lib feeding or more meals isn’t causing less overweight, rather there is a reverse causation: cats that eat less, do not develop overweight.

Based on our univariable analysis, middle-aged, male, domestic shorthair cats were most likely to be overweight, which is in agreement with a previous study [38]. However, our study showed that the size of the average effect of being male and the amount of food on overweight was small.

When a model was built including SPL and age as well as the explanatory variables for overweight, the same direction of effects persisted: a suckling period between 0 to 7 weeks and being middle-age were independently associated with an increased risk for overweight. These effects increased with increasing age of the cat but decreased after 8 year of age. It has been suggested that besides disease, the maintenance energy requirement of older cats increase rather than decrease contributing to the tendency of older cats being underweight if their energy needs are not met [57,58].

In summary, overweight is a frequently encountered condition and can be complex to manage. Our study identified a possible novel and easily modifiable risk factor for overweight, the SPL. Informing cat owners, shelters and cat rescues about the benefits to weaning and separation at an older age—at least 12 weeks—could be an easy and cost-efficient way of improving the quality of life of cats. Based on these study results, it can be suggested that the legal weaning age of 7 weeks should be raised by at least five weeks to ensure a SPL of at least 7 weeks. 

## 5. Limitations

The number of respondents for this study was limited causing low precision for the parameter estimates (i.e. wide confidence intervals). Nevertheless, SPL and age were identified as influencing factors for overweight. The estimated OR have to be interpreted with care but can provide information if the association could be of clinical/biological importance. Respondents were allowed to decide whether or not to participate in the study, which may have led to self-selection bias. However, as most owners are likely to be unaware of the weight-status of their cat, this might be a minor concern. At the end of the survey cat owners were asked whether they wanted to be informed about the fBMI of their cat and all of the owners wanted to know whether their cat was overweight or not, strengthening the idea that most cat owners are unaware of the weight-status of their cat. Even underweight might be hard to determine for cat owners, based on the cat with underweight that was excluded from the study.

The data in this study came from three different clinics in South-Holland. A large proportion of the cats in the study population come from average socioeconomic families. This could introduce selection bias that limits the extent to which the results can be generalized, although two studies showed that there were practically no differences in demography between the owners of overweight and normal cats, gender being the only exception [5,45,48,49,59,60].

The use of a client-owned population made it difficult to quantify food intake and physical activity. The results are thus not free from inevitable imprecision arisen from estimates made by owners. The precise energy content of the food was not calculated.

In this study we used fBMI as a diagnostic tool for obesity instead of BCS. The cut-off point of 30% body fat for overweight might have led to some overrepresentation of overweight cats, since a recent study showed that a BCS of 3/5 (ideal weight) is correlated with a fBMI of 24.6–32.0 [35].

The number of purebred cats of the study sample (29%) exceeded that of the natural population (7%) [29]. It has to be warranted for that although it were mostly purebreds with a SPL of > 12 weeks, the study included 8 domestic shorthairs with a SPL of 13–16 weeks and 2 domestic shorthairs with a SPL of > 16 weeks. There was a significant interaction between purebred and SPL which complicates separating these effects from each other. Part of the breed effect might arise from a longer SPL. A longitudinal cohort study with data solely from breed-matched controls with shorter weaning methods would rule out the breed related bias.

## 6. Conclusions

This study shows that early life nutrition may have detrimental effects on weight status: a predisposition for overweight was found in kittens that suckle 11 weeks or less. These results suggest a longer suckling period might be a simple intervention to improve cat health and well-being, especially for domestic shorthair cats that generally are allowed to suckle 7–8 weeks or less due to the legal separation age.

## Figures and Tables

**Table 1 animals-11-03434-t001:** Inclusion and exclusion criteria for the study animals.

Inclusion Criteria	Exclusion Criteria
Healthy, checked by vet 1–12 years of age Known Suckling Period Length Being first owner or breeder	On a dietetic treatment Food refusal On medication Stray or shelter owned History of food allergy/intolerance Treatments affecting appetite or weight Pregnant, lactating or breeding Bottle-fed as a kitten Fluctuating weight past 6 months Vomiting or diarrhea Impaired mobility Chewing problems Underweight Acutely ill or hospitalized in past 3 months Showing unexplained clinical signs Sick queen Temporary or permanent separation from queen

**Table 2 animals-11-03434-t002:** Univariable analysis of the cats (*n* = 69): Number (%) of cats per demographic factor against overweight status and unadjusted odds ratio with 95% CI*.

Factor	Total		Overweight		Normal Weight			Confidence Interval	
	*n*	% ^1^	*n*	% ^2^	*n*	% ^2^	Odds Ratio	2.5%	97.5%
Total	69	100	34	49.3	35	50.7			
Length of suckling period (weeks)									
0–6	19	27.8	13	68.4	6	31.6	Ref		
7–11	23	33.3	11	47.8	12	52.2	0.42	0.11	1.5
12–16	18	26.1	6	33.3	12	66.7	0.23	0.05	0.88
17–24	9	13.0	4	44.4	5	55.6	0.37	0.07	1.9
Sex									
Male	40	58.0	19	51.7	21	48.3	Ref		
Female	29	42.0	15	47.5	14	52.5	1.2	0.45	3.1
Age (years)									
1.0–3.0	22	31.9	9	40.9	13	59.1	Ref		
3.5–5.0	18	26.1	5	27.8	13	72.2	0.56	0.14	2.1
5.5–7.0	12	17.4	10	83.3	2	16.7	7.2	1.5	56
7.5–12.0	17	24.6	10	58.8	7	41.2	2.1	0.58	7.8
Breed									
Purebred	20	29.0	4	20.0	16	80.0	Ref		
Domestic	49	71.0	30	61.2	19	38.8	6.3	2.0	25
Amount of food									
Less or Prescribed	14	20.3	9	64.3	5	35.7	Ref		
More than prescribed + ad lib	55	79.7	25	45.5	30	54.5	0.46	0.13	1.5
Number of Meals per day									
1–2	35	50.7	20	57.1	15	42.9	Ref		
3–4	12	17.4	5	41.7	7	58.3	0.54	0.13	2.0
Ad lib	22	31.9	9	40.9	13	59.1	0.52	0.17	1.5
Playing with owner									
Daily	27	39.1	11	40.7	16	59.3	Ref		
Not daily	42	60.9	23	54.8	19	45.2	1.8	0.7	4.8
Free Outdoor access									
Yes	35	50.7	20	57.1	15	42.9	Ref		
No	34	49.3	14	41.2	20	58.8	0.53	0.20	1.4

* The full model started with variables: length of suckling period + age + sex + amount of food + number of daily meals per day + playing with owner + free roaming. Abbreviations: *n* = Number; Ref = Reference category; ^1^ Percentage of category within the variable (column Total); ^2^ Percentage of overweight status within the category of the variable (row).

**Table 3 animals-11-03434-t003:** Final multivariable model * of the cats (*n* = 69) with adjusted odds ratios and 95% confidence intervals for overweight.

Variable		Confidence Interval	
	Odds Ratio	2.5%	97.5%
Length of suckling period (weeks)			
0–6	Ref		
7–11	0.45	0.11	1.8
12–16	0.31	0.07	1.3
17–24	0.23	0.03	1.4
Age (years)			
1.0–3.0	Ref		
3.5–5.0	0.59	0.14	2.3
5.5–7.0	8.0	1.5	69
7.5–12	2.1	0.54	8.6

* The full model started with variables: length of suckling period + age + sex + amount of food + number of daily meals per day + playing with owner + free roaming; Abbreviations: Ref = Reference category.

**Table 4 animals-11-03434-t004:** Number (%) of purebred and domestic shorthair cats per suckling period group and demographic factor.

Factor	Total		Purebred		Domestic Shorthair		
	*n*	% ^1^	*n*	% ^1^	*n*	% ^1^	*p*-Value ^2^
Total	69	100	20	29.0	49	71.0	
Suckling Period Length (weeks)							<0.001
0–6	19	27.5	1	5.0	18	36.7	
7–11	23	33.3	2	10.0	21	42.9	
12–16	18	26.1	10	50.0	8	16.3	
> 16	9	13.0	7	35.0	2	4.1	
Weight status							0.003
Overweight	34	49.3	4	20.0	30	61.2	
Non-overweight	35	50.7	16	80.0	19	38.8	
Sex							0.59
Male	40	58.0	13	65.0	27	55.1	
Female	29	42.0	7	35.0	22	44.9	
Age (years)							0.009
1.0–3.0	22	31.9	2	10.0	20	40.8	
3.5–5.0	18	26.1	10	50.0	8	16.3	
5.5–7.0	12	17.4	2	10.0	10	20.4	
7.5–12.0	17	24.6	6	30.0	11	22.4	
Amount of food							0.74
Less or recommended	14	20.3	3	15.0	11	22.4	
More than recommended	55	79.7	17	85.0	38	77.6	
Number of Meals per day							0.008
1–2	35	50.7	6	30.0	29	59.2	
3–4	12	17.4	2	10.0	10	20.4	
Ad libitum	22	31.9	12	60.0	10	20.4	
Playing with owner							1.0
Daily	27	39.1	8	40.0	19	38.8	
Not daily	42	60.9	12	60.0	30	61.2	
Free Outdoor access							0.008
Yes	35	50.7	5	25.0	30	61.2	
No	34	49.3	15	75.0	19	38.8	

Abbrevations: *n* = Number; ^1^ Percentage of category within the variable (column); ^2^ Fisher exact test.

## Data Availability

The data sets analyzed in the present study are available from the corresponding author on reasonable request.

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
