# Peer review of "Kittens That Nurse 7 Weeks or Longer Are Less Likely to Become Overweight Adult Cats"

_animals, 2021, doi:10.3390/ani11123434_

Round 1
Reviewer 1 Report
This is a very interesting study which holds some fascinating potential to influence obesity in pet cats. My main questions are around the analysis and the exclusion of breed from the logistic regression even though it was shown to be important. More below. I also have some specific comments throughout.
Introduction: in general, please clarity which references are for cats (vs pigs or humans). It isn’t consistently clear without checking the reference list.
Line 80-1: for the objective, a bit more clarity about the cat population (some purebreds some not, different sources, etc.) would help here.
Lines 83-86: I think this is more results. If it is determined it should remain in methods, please move to line 132. Otherwise, it isn’t clear that this is the summary AND that many of the cats weren’t patients at the clinics but were seen there for the study.
Lines 87-9: these are the included 96 cats. But later on, in the manuscript (end of the results section) we find that some were actually recruited directly from the clinics? Please clarify here and there. So, some were already patients there?
Line 96: please add: …criteria were met, an appointment was scheduled at one of the three participating clinics”, assuming that this is correct.
Line 120-2: There is some confusion here. I believe that there were cats with restricted outdoor access, those with no outdoor access and those with unrestricted (is that in terms of time or a cat door?) access. And then there was a question about the frequency of playing with the owner. Please edit this section for completeness and clarity and be sure the wording is carried into the tables.
Tables: please round percentages to whole numbers for ease of comprehension. Please round OR/CI to 2 significant digits (so 1.26 become 1.3; 13.13 becomes 13). For the 1 and 2 subscripts I think it will be clearer is if you label 1 as the percentage down that column and 2 as the percentage across the row. Making the variable column left justified makes it easier to read and see what is the main variable and which are the categories within that variable. Same comments for the supplemental tables.
Line 153: 96 cats met the eligibility requirements; how many were screened out? Anything that can be said about the most common reasons for not meeting the requirements? That provides some info on potential selection bias and should be mentioned as a limitation in the discussion.
Section 3.2: I do appreciate table 3. As I indicated above, I’m not sure why breed wasn’t included in the multivariable model so that should be included in the manuscript—I’m guessing maybe there weren’t enough observations to include and the model wouldn’t run? But unless it was impossible, it should be included. And an assessment of whether it was a confounding variable or part of an interaction term should also be included. Another possible way to explore this is to run two separate modeling efforts, one for purebreds and one for mixed breed cats. Then see how similar they are in significant variables and magnitude/direction of the odds ratios.
Table 3: it looks like the variable for overweight has row percentages rather than column percentages? Please double check and edit. I’d also like to see a statistical comparison for each variable by pure vs mixed breed and have the p-value included in this table. That stats analysis will also need to be added to the methods section.
Table 4: was there a statistically significant interaction between age and SPL? Was there enough statistical power to run that analysis?
Line 203-5: Because most of the data set consisted of the cats from the clinics, I would expect that subset of data, when analyzed, to strongly resemble the full analysis—which is what was found. What would be more helpful is to do the analysis for only the cats from the other recruitment sources. Then if those are similar to the full analyses there is a stronger argument to be able to combine them. Source of recruitment could also be included in the multivariable model to control confounding. One could also do a univariable analysis of recruitment type and obesity and recruitment type and SPL and identify any relationships that way.
Line 219: actually the 5.5-7 year category isn’t much different from the oldest age category (and the OR are similar and within each other’s 95% CI) so that I think it is more of a threshold where risk increases about 4 fold after 5 years of age (once SPL is accounted for).
Line 245: and of course, genetic differences could be involved. Please add to manuscript.
Line 246-7: did the study have enough statistical power to detect a difference given the findings? Please address this.
Paragraph starting line 258: I suspect that the cats who get treats and many meals or ad libitum could be a result of cats who aren’t big eaters or who are considered to be picky. And therefore, ad lib or many meals isn’t causing less overweight, rather there is reverse causation. We don’t know but that possibility should be stated in the text along with any other possible explanations.
Line 286: how was the study presented to potential participants? As one about obesity? Kittenhood? That can influence who decides to participate. Please clarify in the methods and comment here.
Supplemental tables 3 and 5: I’m not clear how these are different from the main table in the manuscript except for where ad libitum is included for table 3.
Author Response
We would like to thank the reviewers for taking the time to assess our manuscript. All the concerns the reviewers raised have been addressed. In this document you will find the line numbers in red referring to these additions.
Reviewer 1
This is a very interesting study which holds some fascinating potential to influence obesity in pet cats. My main questions are around the analysis and the exclusion of breed from the logistic regression even though it was shown to be important. More below. I also have some specific comments throughout.
- Introduction: in general, please clarity which references are for cats (vs pigs or humans). It isn’t consistently clear without checking the reference list – We have made adjustments throughout the introduction and hope it is clear now which references are for which study objects.
- Line 80-1: for the objective, a bit more clarity about the cat population (some purebreds some not, different sources, etc.) would help here. We have added additional information about the sample in lines (93-104)
- Lines 83-86: I think this is more results. If it is determined it should remain in methods, please move to line 132. Otherwise, it isn’t clear that this is the summary AND that many of the cats weren’t patients at the clinics but were seen there for the study. Adjusted as requested within materials and methods.
- Lines 87-9: these are the included 96 cats. But later on, in the manuscript (end of the results section) we find that some were actually recruited directly from the clinics? Please clarify here and there. So, some were already patients there? That is correct and we have added additional information in lines (93-104)
- Line 96: please add: …criteria were met, an appointment was scheduled at one of the three participating clinics”, assuming that this is correct. This is correct, why this sentence was added in lines (102-104)
- Line 120-2: There is some confusion here. I believe that there were cats with restricted outdoor access, those with no outdoor access and those with unrestricted (is that in terms of time or a cat door?) access. And then there was a question about the frequency of playing with the owner. Please edit this section for completeness and clarity and be sure the wording is carried into the tables. We agree and have clarified this part in lines (129-132)
- Tables: please round percentages to whole numbers for ease of comprehension. Please round OR/CI to 2 significant digits (so 1.26 become 1.3; 13.13 becomes 13). For the 1 and 2 subscripts I think it will be clearer is if you label 1 as the percentage down that column and 2 as the percentage across the row. Making the variable column left justified makes it easier to read and see what is the main variable and which are the categories within that variable. Same comments for the supplemental tables. We have optimized the figures as suggested for more clarity
- Line 153: 96 cats met the eligibility requirements; how many were screened out? Anything that can be said about the most common reasons for not meeting the requirements? That provides some info on potential selection bias and should be mentioned as a limitation in the discussion. Thank you for pointing this out. Information about the excluded cats has been added and can be found in lines (162-164)
- Section 3.2: I do appreciate table 3. As I indicated above, I’m not sure why breed wasn’t included in the multivariable model so that should be included in the manuscript—I’m guessing maybe there weren’t enough observations to include and the model wouldn’t run? But unless it was impossible, it should be included. That is correct, there were not enough observations to be able to run the model.
- And an assessment of whether it was a confounding variable or part of an interaction term should also be included. Another possible way to explore this is to run two separate modeling efforts, one for purebreds and one for mixed breed cats. Then see how similar they are in significant variables and magnitude/direction of the odds ratios. Although we do appreciate the suggestion, this is not possible because there are no purebreds with a SPL <12 weeks. Breeder associations demand from breeders that want their pedigree registered that kittens are not adopted out before 12 weeks. The breeders associations take disciplinary action against members who do not comply with the breeders guidelines.
- Table 3: it looks like the variable for overweight has row percentages rather than column percentages? Please double check and edit. I’d also like to see a statistical comparison for each variable by pure vs mixed breed and have the p-value included in this table. That stats analysis will also need to be added to the methods section. We have changed the analysis and data descriptions taken your comments into account (Table 4)
- Table 4: was there a statistically significant interaction between age and SPL? Was there enough statistical power to run that analysis? No, there were no other significant interactions, number of animals per group was a limitation which is now mention more clearly in the discussion
- Line 203-5: Because most of the data set consisted of the cats from the clinics, I would expect that subset of data, when analyzed, to strongly resemble the full analysis—which is what was found. What would be more helpful is to do the analysis for only the cats from the other recruitment sources. Then if those are similar to the full analyses there is a stronger argument to be able to combine them. Source of recruitment could also be included in the multivariable model to control confounding. One could also do a univariable analysis of recruitment type and obesity and recruitment type and SPL and identify any relationships that way. Thank you for the insightful comment. We have chosen to omit the > 16 week recruitment subset.
- Line 219: actually the 5.5-7 year category isn’t much different from the oldest age category (and the OR are similar and within each other’s 95% CI) so that I think it is more of a threshold where risk increases about 4 fold after 5 years of age (once SPL is accounted for). In the geriatric group, the range is different, that’s why we would like to keep it as it was.
- Line 245: and of course, genetic differences could be involved. Please add to manuscript. That is a fair remark. This was added in lines (295-296).
- Line 246-7: did the study have enough statistical power to detect a difference given the findings? Please address this. This is now discussed in the limitations section
- Paragraph starting line 258: I suspect that the cats who get treats and many meals or ad libitum could be a result of cats who aren’t big eaters or who are considered to be picky. And therefore, ad lib or many meals isn’t causing less overweight, rather there is reverse causation. We don’t know but that possibility should be stated in the text along with any other possible explanations. This theory was added in lines (324-327)
- Line 286: how was the study presented to potential participants? As one about obesity? Kittenhood? That can influence who decides to participate. Please clarify in the methods and comment here. More information about this can be found in lines (98-100) Supplemental tables 3 and 5: I’m not clear how these are different from the main table in the manuscript except for where ad libitum is included for table 3. We have omitted tables 3 and 5 from the supplemental material.
We would like to thank the referees again for taking the time to review our manuscript.

Reviewer 2 Report
The authors present a study examining the potential for suckling period length (SPL) as a predictor of excessive weight gain in adulthood in domestic cats owned as pets. They seek to address this aim through data gathered via a questionnaire administered to cat owners regarding their cat’s history, age, feeding habits, exercise and home environment. The study is simple and provides an interesting view on the issue of overweight cats as a clinical problem as well as a potentially simple solution to the issue at hand. However, the study suffers from one major problem: the SPL and the breed status of the cats co-vary and this association is not adequately accounted for in the study design or the analyses.
Throughout the authors highlight the confounding effect of breed and SPL being associated. It seems odd that the authors would be very obviously aware of this effect yet not choose to address it through the simple separation of the cats according to breed status (i.e. domestic shorthair versus purebred) in the analyses. By analysing the two groups independently, the analysis would be far cleaner and the interpretation of the results would not be fraught with the complications of trying to disentangle SPL from breed effects. This is particularly confusing when the authors outline all of the evidence in the literature to suggest that the purebred group differs from the domestic shorthair group in a myriad of ways which are not accounted for by the questionnaire data. Presumably the groups were combined in the analyses in order to bolster the apparent sample size but in doing so the authors undermine the robustness of their own results. It is my strong recommendation that the authors separate out the groups in order to achieve a clearer picture of the relationships they seek to describe.
Otherwise, the submission is acceptable and clear for the most part. There are a few confused uses of specific terms (e.g. gender) and the English language could be improved through proof-reading by a native English speaker. The overall structure of the discussion is somewhat confusing. The logical flow is not clear and many of the ideas presented, while valid, appear to be unordered and random. For example, lines 223-224 seem to be the primary finding of the study and thus, I would expect that they would be mentioned upfront before the other findings. However, they appear three paragraphs into the discussion and are at the end of the paragraph in question.
With the provision that the authors restructure the analysis of the data to control for the confounding effects of breed status and SPL, it is my opinion that the submission would be acceptable for publication. It is with this in mind that I recommend that the submission undergo minor revision for reconsideration.

Author Response
We would like to thank the reviewers for taking the time to assess our manuscript. All the concerns the reviewers raised have been addressed. In this document you will find the line numbers in red referring to these additions.
Reviewer 2
The authors present a study examining the potential for suckling period length (SPL) as a predictor of excessive weight gain in adulthood in domestic cats owned as pets. They seek to address this aim through data gathered via a questionnaire administered to cat owners regarding their cat’s history, age, feeding habits, exercise and home environment. The study is simple and provides an interesting view on the issue of overweight cats as a clinical problem as well as a potentially simple solution to the issue at hand. However, the study suffers from one major problem: the SPL and the breed status of the cats co-vary and this association is not adequately accounted for in the study design or the analyses.
Throughout the authors highlight the confounding effect of breed and SPL being associated. It seems odd that the authors would be very obviously aware of this effect yet not choose to address it through the simple separation of the cats according to breed status (i.e. domestic shorthair versus purebred) in the analyses. By analysing the two groups independently, the analysis would be far cleaner and the interpretation of the results would not be fraught with the complications of trying to disentangle SPL from breed effects. This is particularly confusing when the authors outline all of the evidence in the literature to suggest that the purebred group differs from the domestic shorthair group in a myriad of ways which are not accounted for by the questionnaire data. Presumably the groups were combined in the analyses in order to bolster the apparent sample size but in doing so the authors undermine the robustness of their own results. It is my strong recommendation that the authors separate out the groups in order to achieve a clearer picture of the relationships they seek to describe.
Otherwise, the submission is acceptable and clear for the most part. There are a few confused uses of specific terms (e.g. gender) and the English language could be improved through proof-reading by a native English speaker. The overall structure of the discussion is somewhat confusing. The logical flow is not clear and many of the ideas presented, while valid, appear to be unordered and random. For example, lines 223-224 seem to be the primary finding of the study and thus, I would expect that they would be mentioned upfront before the other findings. However, they appear three paragraphs into the discussion and are at the end of the paragraph in question
With the provision that the authors restructure the analysis of the data to control for the confounding effects of breed status and SPL, it is my opinion that the submission would be acceptable for publication. It is with this in mind that I recommend that the submission undergo minor revision for reconsideration.
We thank the reviewer for these constructive remarks and agree. We have chosen to omit the > 16 week recruitment subset and did a new analysis with just the ‘clinic animals’. Furthermore, we have changed “gender” into “sex” throughout the document. With regards to the discussion, we have chosen to change the order in such a way that major findings are reported, followed by other findings, limitations, and summary. But we agree the text needed editing for clarity and logic. We have revised the discussion to address your concerns and hope that it is now clearer.
We would like to thank the referees again for taking the time to review our manuscript.
